# An Innovative Sorption Technology for Removing Microplastics from Wastewater

**Marketa Spacilova** [1,*], **Pavel Dytrych** [1], **Martin Lexa** [2], **Lenka Wimmerova** [3], **Pavel Masin** [4], **Robert Kvacek** [5] **and Olga Solcova** [1]

1   Department of Catalysis and Reaction Engineering, Institute of Chemical Process Fundamentals of the CAS, Rozvojova 135, 165 00 Prague, Czech Republic
2   Department of Wood Processing and Biomat, Faculty of Forestry and Wood Sciences, Czech University of Life Sciences Prague, Kamycka 129, 165 00 Prague, Czech Republic
3   Department of Applied Ecology, Faculty of Environmental Sciences, Czech University of Life Sciences Prague, Kamycka 129, 165 00 Prague, Czech Republic
4   DEKONTA a.s., Dretovice 109, 273 42 Dretovice, Czech Republic
5   Prazske Vodovody a Kanalizace, a.s., Ke Kablu 971, 102 00 Prague, Czech Republic
*   Correspondence: spacilova.marketa@icpf.cas.cz; Tel.: +420-220-390-280

**Abstract:** This study is focused on technology development for microplastic removal from wastewater using a sorption process, which would be suitable not only as a tertiary stage of purification in wastewater treatment plants but also for other waters, e.g., process water and surface water. Therefore, cheap natural materials such as zeolites and bentonites were tested as possible sorbents. This study aims not only at sorbent selection but also at their possible modification by a special water regime improving sorption efficiency and lifetime. Microplastic particles of the majority of common types of plastics were prepared by a newly developed abrasion technique from various plastic items used at home, thus microplastic particle sizes and shapes corresponded to the real microplastics found in waters. Based on results with high reproducibility, a novel method for microplastic characterization based on Raman spectroscopy in combination with SEM/EDX was developed. The removal of microplastics from waste water was tested not only at the laboratory scale but also in a developed semi-operational sorption unit at a real wastewater treatment plant throughout the year with the efficiency of over 90%.

**Keywords:** microplastics; water treatment; sorption; scale up; bentonite

## 1. Introduction

Microplastics in water are among the most problematic emerging pollutants and have become a major environmental issue. It is estimated that the leakage of secondary microplastics alone reaches 176,000 tons per year in European surface waters [1]. In an effort to prevent the release of microplastics into the environment, the European Chemicals Agency (ECHA) prepared a proposal to limit microplastics in products placed on the EU market in January 2019 [2], and in August 2022, the ECHA further prepared a proposal to change the list of substances subject to restrictions according to Annex XVII of the REACH Regulation [3]. Owing to these bans, it is expected that 500,000 tons of microplastics will not be released into the environment over the next twenty years. Their adoption will prevent the entry of primary microplastics; however, secondary microplastics will continue to occur in the environment and will need to be removed from the environment, not just water.

Ubiquitous microplastics are ingested by aquatic creatures from microorganisms to fish (for Mullus barbatus [4] and, at the end of the food (for radish, see [5]) chain, by humans [6–12]. Four high-production-volume polymers applied in plastic (polyethylene terephthalate, polyethylene, polystyrene derivates as expanded polystyrene and acetonitrile butadiene styrene) were identified and quantified for the first time in blood [13]. Dumping

of plastic waste, the lack of standard detection methods with specific removal techniques together with the slow disposal rate of microplastics make them ubiquitous in the environment [14]. There are some biological, chemical, electrochemical and physical techniques for microplastic removal; however, their wide applicability and cost-effectiveness are issues. Recently, Sing et al. [15] discussed the existing and upcoming treatment technologies for microplastic removal from wastewater and mentioned that conventional wastewater treatment plants are not fully efficient. In fact, the final effluent contains significant amounts of microplastics. Sun et al. [16] and Liu et al. [17] reviewed the characteristics and removal of microplastics in 38 wastewater treatment plants (WWTPs) in 11 countries worldwide and reported that 88% overall microplastic removal could be achieved when tertiary treatment was absent, and 97% when tertiary treatment was involved. It follows that it is necessary to equip WWTPs with tertiary treatment; however, it is also necessary to consider the economic factor.

Various removal methods for microplastics exist; nevertheless, they are, unfortunately, only verified at the laboratory scale [18–20]. Coagulation with aluminum and iron salts is widely used in water treatment facilities; however, process efficiency strongly depends on the coagulant type and microplastic size/type and is generally low (up to 50%). However, under optimized conditions and with high coagulant doses, high efficiency can be reached (up to 91%) [21]. Electrocoagulation also shows an equally similar efficiency [22]. Biochar or activated carbon filtration appears quite effective; however, the process is slow, results in pore clogging, and regeneration is difficult [23]. In addition, activated carbon filtration has been tested together with coagulation [15,24,25]. The authors mentioned other methods, such as sorption on algae, bio-inspired molecules, metal organic framework (MOF) foams or photocatalytic micromotors; regrettably, the majority of them cannot be used for practical purposes.

Interestingly, Ariza-Tarazona et al. [26] found that exposure to photocatalysis (e.g., with $TiO_2$) also decreased the microplastic concentration, perhaps indicating microplastic photocatalytic degradation. However, the application of these techniques might not currently be feasible owing to the high costs and complexities involved; nevertheless, in the near future and considering the necessary degradation of emerging pollutants, such techniques will have to be employed. In this context, the development of smaller filter units as a tertiary treatment, which would remove microplastics and other pollutants, appears to be a suitable approach [15].

In fact, there is no operational-scale microplastic removal technology in operation.

Membrane filtration [27,28] and coagulation–flocculation–settling [29] treatments are non-destructive, requiring an additional step to degrade microplastics. Biological treatment was demonstrated as unsatisfactory for microplastic treatment. On the other hand, a few recent works identify advanced oxidation processes (photocatalysis, Fenton, and wet oxidation) as feasible alternatives, since they present high-efficiency microplastic degradation ($\approx$30–95%) [30–32]. However, further studies should be conducted to evaluate the performance of advanced oxidation processes on the degradation of microplastics in real conditions. Tests in larger treatment systems are critical to promote a scale up for the real application [33]. A very limited number of studies on microplastic degradation technologies are available in the research literature. Clearly, a significant gap in knowledge is the lack of results from scale-up studies as well as laboratory studies with the potential for a scale up [34].

For this reason, our study is focused on the development of a technology for microplastic removal from water using a sorption process, which would be suitable not only as a tertiary stage in wastewater treatment plants but also as a secondary stage of purification of surface and industrial waters. Due to economic reasons, this study utilizes low-cost natural materials such as zeolites or bentonites and their possible modification with the aim to increase removal efficiency and their lifetime, not only at the laboratory scale but also in a real waste water treatment plant.



## 2. Materials and Methods

### 2.1. Used Materials

A list of the used sorbents and plastic materials is presented in Tables 1 and 2, including their description and source. Particles sizes were specified directly by the manufacturer, and the sizes of the prepared microplastic particles were determined by scanning electron microscopy (SEM, Tescan Indusem).

**Table 1.** Used adsorbents.

| Material Used | Description | Origin/Manufacturer |
|---|---|---|
| Bentonite Braňany EXTRA | Clay, powder, particle size < 3 mm | Dekonta, a.s. |
| Bentonit Braňany STELIVO | Clay, powder, particle size < 3 mm | Dekonta, a.s. |
| Zeolite Clinoptilolite | Zeolite, particle size 1–2.5 mm | Uprav vodu.cz |

**Table 2.** Used plastic materials.

| Used plastic Material | Description | Origin/Manufacturer |
|---|---|---|
| Polyethyleneglycol terephthalate | Particle size < 500 μm | Drinking bottle |
| Polypropylene | Particle size < 500 μm | A cup of yogurt |
| Polystyrene | Particle size < 500 μm | A cup of yogurt |
| Polycarbonate | Particle size < 500 μm | CD/DVD disc |
| Polymethyl methacrylate | Particle size < 500 μm | Raw piece of Plexiglas |
| Stainless steel sieve | Wire stainless steel fabric with mesh size 25, 50, 500 μm | Euro SITEX s.r.o. |

Textural characteristics of specific surface area ($S_{BET}$), specific surface area of mesopores ($S_{meso}$), and pore size distribution were determined by physical adsorption of nitrogen at $-196\ °C$ on ASAP 2050 and 2020 instruments (Micromeritics). The pretreatment preceding the analysis consisted of drying the samples under vacuum (1 Pa) at 105 °C for 12 h. The micropore volume ($V_{mikro}$), $S_{BET}$ and $S_{meso}$ were determined from the adsorption–desorption isotherm using a modified BET (according to S. Brunauer, B.H. Emmet and E. Teller; 1938) equation [35] and a t-plot using the standard Lecloux–Pirard isotherm [36].

### 2.2. Preparation and Characterization of Microplastics

Microplastics for laboratory tests were prepared by simulated abrasion. The preparation consisted of abrasion by sandpaper, a corundum grinding stone and a direct electric grinder. The force and preparation time of particles depended on the properties of the individual plastic materials (stiffness, softness, and brittleness).

Prepared microplastic particles were characterized by scanning electron microscopy (SEM, Tescan Indusem). The images were taken at the accelerating voltage of 15 kV. The surface of microplastic particles was mapped by SEM, which enabled the determination of their size and shape.

The particles were characterized by Raman spectroscopy (Nicolet Almega XR with Olympus BX51 microscope, excitation laser 473 nm, 5 mW power). The library for microplastic detection was created from spectra of the individual microplastics prepared by abrasion.

Elemental analyses were performed by SEM equipped with energy-dispersive X-ray spectroscopy (EDX, XFlash 5010 detector and Quantax 200). All measurements were performed at an accelerating voltage of 15 kV. The combination of EDX and micro-FTIR (Fourier transform infrared spectrometer, Nicolet Avatar 360, Zn Se ATR) methods was used for a complete qualitative analysis of samples containing multiple types of microplastics.

### 2.3. Laboratory Sorption Experiments

The sorption apparatus consisted of a 50 cm-long glass tube with an inner diameter of 4.1 cm, which was terminated by a tap; see Figure 1. Above the tap, the tube was

provided with a perforated glass partition supported with a stainless sieve (mesh 1 mm). The sorption part consisted of a 2 cm layer of clinoptilolite covered by a sorbent bentonite (Branany EXTRA). The sorbent height ranges from 1 to 20 cm according to the experiment.

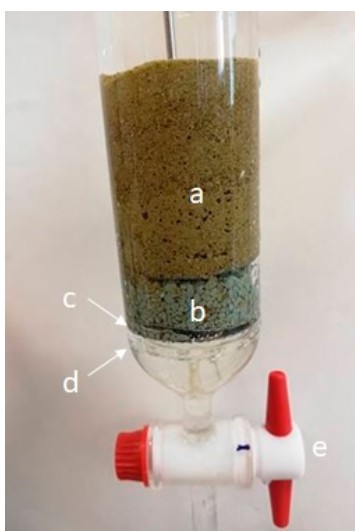

**Figure 1.** Apparatus with the sorbents bed; (a) layer of bentonite Braňany EXTRA; (b) layer of clinoptilolite; (c) stainless sieve (mesh 25 µm); (d) glass perforable bottom; (e) tap.

After filling with the sorbent, the column was flushed with at least two liters of water, which ensured a constant flow of water. The capture of microplastics at the laboratory scale was tested using simulated samples with a volume of 0.5 L tap water. The water was running off the tap for 5 min. It was used to simulate potable water. The pH was tested once a week with small differences found (6.6 + −0.2). The salinity was not tested. It contained 5–50 mg of microplastics. A stainless steel sieve with a mesh size of 25 µm was placed at the exit of the column, on which microplastics passing through the column were captured. Sampling was carried out after a flow of 0.5 L of individual samples containing the prepared microplastics. Following that, the column was washed with 0.5, 1, 1.5, 2.5 and 10 L of clean water. The flow rate depended on the amount of sorbent in the column. The captured microplastics on the stainless steel sieve were analyzed by SEM combined with EDX and FTIR. This combination of methods made it possible to evaluate not only the amount but also the type of microplastic. The images of individual microplastic particles captured on the sieve were obtained using SEM, and spectra characteristic for individual types of microplastic were determined using EDX and FTIR, which made it possible to determine the identity of the captured microplastic particles.

Based on these experiments, the semi-operational column presented in the Results section was developed and constructed (see Results section).

*2.4. Sampling and Analysis by Raman Spectroscopy*

Wastewater samples were taken through a sampling device containing three stainless steel filters with a diameter of 17 mm and metal fabric roughness of 500, 50 and 25 µm (Eurositex, Czech Republic). The sampling device was originally designed for pilot-scale testing; however, initially, it was inspired by Martin et al. [37].

Before sampling, each filter was washed twice in ultrapure water (PURELAB flex 1, ELGA LabWater, High Wycombe, UK) and ethanol p.a. (Sigma-Aldrich, Prague, Czech Republic), dried in a laboratory drier at 45 °C (ED 115, Binder, Tuttlingen, Germany) and placed into by the same procedure prewashed 50 mL glass bottle sealed with a lid with internal aluminum foils. After sampling, each filter was taken from the sampling device and placed into the same bottles sealed and tightly closed. Subsequently, and on the same day, the samples were transported into an analytical laboratory.

In the laboratory, each filter was taken out from its glass bottle by a stainless steel tweezers washed in ethanol p.a. and put into a Petri dish of 40 mm diameter, prewashed by the same procedure as the sample's glass bottle. All filters were dried at 45 °C for 24 h prior to their analyses on a Raman device (WiTec WMT50 with alpha300 R microscope, WiTec, Ulm, Germany). On each filter, a composite image of the area of 1 cm$^2$ (resolution 1.25 µm/px) was taken at 100 times magnification and sharpening of the Z-axis. Within this area, particles were searched. Each particle was first photographed at five hundred times magnification (resolution 0.25 µm/px), and then its spectrum was measured using a Raman laser excitation laser of 532 nm.

### 2.5. Determination of the Effectiveness of Removal

To determine the degree of microplastic removal at the technology/laboratory scale, a statistical methodology was developed to evaluate the Raman spectra and determine the frequency/concentration of microplastics. The results were analyzed in R-language using integrated statistical packages (StatModel). Preliminary concentrations of microplastics in WWTP water and laboratory samples were compared using an (unpaired) two-sample *t*-test [38]. Particle size distributions were fitted to "bin" widths of 25–50–500 µm. Furthermore, the results of visual analysis of microplastics and Raman spectroscopy were compared using Spearman's correlation at both the laboratory and industrial scales.

## 3. Results and Discussion

First, it was inevitable to have sufficient quantities of different types of microplastics for laboratory experiments with simulated water. Microplastic particles found in real water possess all possible shapes. Unfortunately, commercially available microplastic particles exist only in the form of balls. Thus, the first task was focused on the preparation of microplastic particles, whose shape would correspond to real microplastics.

### 3.1. Preparation and Characterization of Microplastic Particles

Several options of microplastic particle preparation by various abrasion tools were tested. Initially, sandpaper with different grain sizes was applied. The obtained particles precisely corresponded to the requirements related to sizes and shapes; regrettably, the particle amount for the majority of plastics was nearly negligible. The only particles possible to be prepared by means of using sandpaper were microplastic particles of polytetrafluoroethylene (PTFE) and polyethylene terephthalate (PET). The same results were obtained with a grinding stone. Thus, an electric straight grinder with a rich constellation of diamond bits was applied. The obtained results were excellent. Prepared microplastic particles of all types of plastics could be prepared in a sufficient amount, and the particle sizes and shapes corresponded to the real microplastics found in waters. The shape and particle sizes of four types of microplastics are shown in Figure 2. The microplastic particles prepared from polyethylene glycol terephthalate (PET bottles) are shown in Figure 2a. This material was soft; thus, very small particles below 40 µm were formed together with larger irregular particles with a high degree of fragmentation (>100 µm). The particle sizes of polyterrafluoroethylene (PTFE) prepared from the raw piece of PTFE rod, which varied between 60 and 100 µm, are shown in Figure 2b.

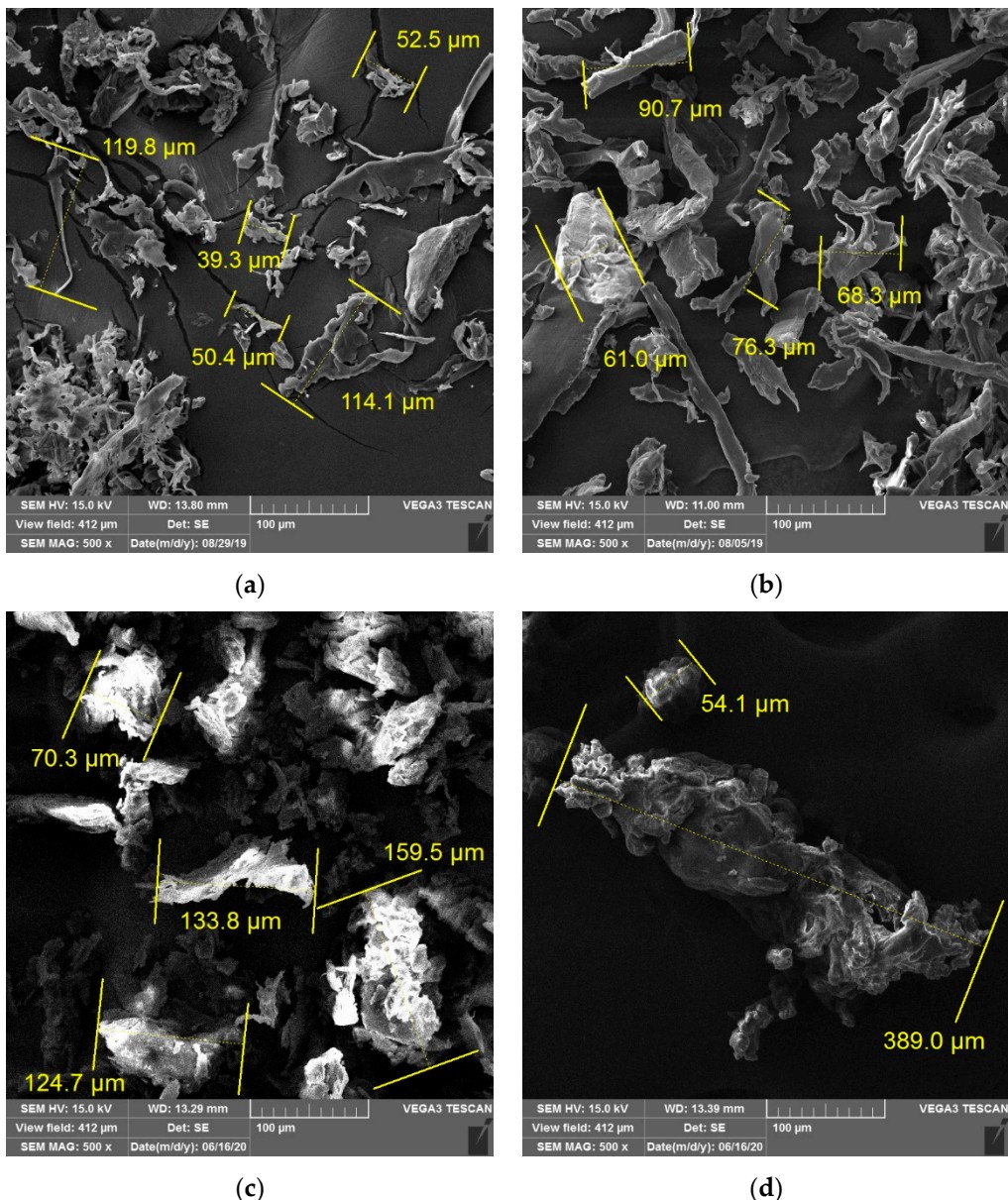

**Figure 2.** Images of individual microplastics taken by SEM: (**a**) PET SP 320; (**b**) PTFE SP 240; (**c**) PS EG 400; (**d**) PC EG 400.

Polystyrene particles (PS) prepared from cups (minimally of 100 μm in size) are shown in Figure 2c. Polycarbonate (PC) particles prepared from CD/DVD discs are shown in Figure 2d. On the other hand, the polycarbonate was hard, and there was partial heating during grinding; therefore, these particles revealed the highest sizes from approximately 50 to 400 μm.

### 3.2. Sorbent Selection and Modification

Since the work was focused on a scale-up process suitable for real use at WWTPs, properties and price were taken into consideration when choosing a suitable sorbent. Thus, three sorbents—zeolite clinoptilolite and two types of bentonites, Branany EXTRA and STELIVO—were chosen. Their textural characteristics are summarized in Table 3 and Figures 3 and 4.

**Table 3.** Characterization of sorbents.

| Sorbent | $S_{BET}$ (m²/g) | $S_{meso}$ (m²/g) | $V_{tot}$ (mm³$_{liq}$/g) | $V_{micro}$ (mm³$_{liq}$/g) |
|---|---|---|---|---|
| Bentonite Branany EXTRA | 107 | 76 | 157 | 17 |
| Bentonite Branany STELIVO | 86 | 58 | 113 | 13 |
| Zeolite Clinoptilolite | 33 | 21 | 114 | 6 |

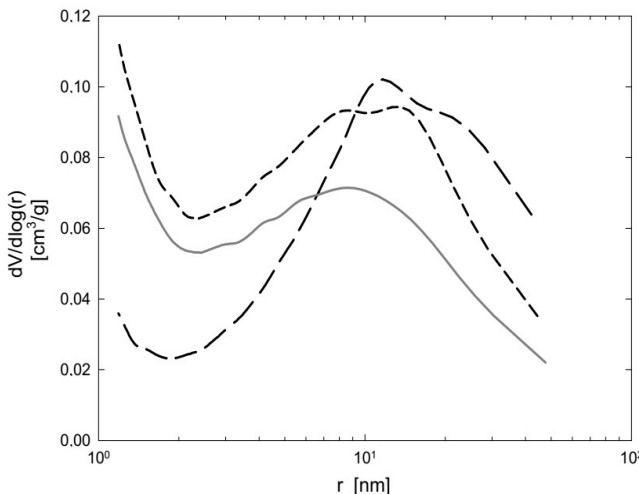

**Figure 3.** Pore size distribution from the adsorption branch of the isotherm: ——— zeolite clinoptilolite; - - - - Bentonite Branany EXTRA and —— Bentonite Branany STELIVO.

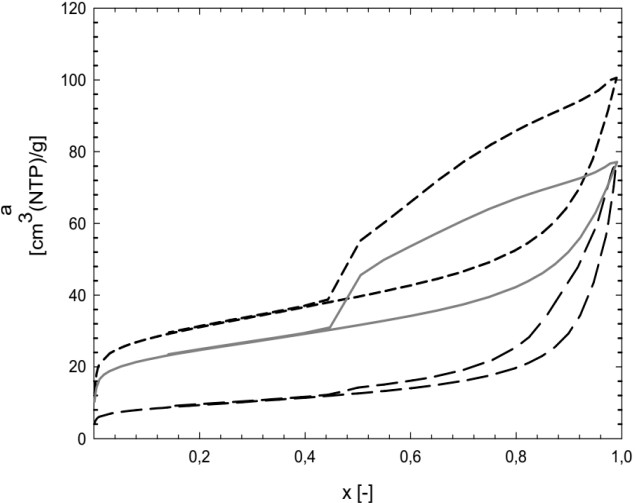

**Figure 4.** The adsorption isotherms of used sorbents: ——— zeolite clinoptilolite; - - - - Bentonite Branany EXTRA and —— Bentonite Branany STELIVO.

The bentonites Branany EXTRA and STELIVO revealed similar values of $S_{BET}$ 107, resp. 86 m²/g, as well $V_{micro}$ 17, resp. 13 mm³$_{liq}$/g, which were significantly higher than for zeolite clinoptiolite. Zeolite clinoptiolite and Bentonite Branany STELIVO possessed the same total pore volume of 113–114 mm³$_{liq}$/g. However, the Bentonite Branany EXTRA possessed the highest values of all measured characteristics. Moreover, Bentonite Branany EXTRA contained a dominant proportion of montmorillonite, which enabled the intercalation of plastic microparticles into individual planar layers. For that reason, the natural sorbent Bentonite Branany EXTRA was chosen for subsequent experiments.

At first, this sorbent was partially dried and grounded in a ball mill to particles below 3 mm in size. Bentonite treated in this way could not be used directly regarding the large

amount of dust particles that clogged the apparatus. Sieving did not work, as bentonite was very hygroscopic, and there were clumps of particles in which many dust particles were absorbed. As another possibility, the hydrodynamic separation of small bentonite particles that would remain suspended in contact with water was tested.

Raw bentonite was washed with ten times the amount of water. Following the washing, the large particles sedimented, while the dust particles remained dispersed in the water. In approximately 20–60 s, the dust particles were removed by decantation. This process was repeated (6 to 10 cycles), while the used water containing fine particles of bentonite was recycled during the process by filtration.

Based on the image analysis of the modified bentonite, a geometric mean particle diameter of approximately 150 μm was obtained; however, a significant part of the population contained particles of even above 250 μm. The entire hydrodynamic process was reproducibly repeated at least five times. The dispersion of particle sizes depended on the degree of flushing (number of cycles and mass fraction of the fine fraction in raw bentonite). Figure 5a shows the bentonite particles before the modification, and Figure 5b after the hydrodynamic treatment, within which the large particles were fragmented and a more homogeneous mixture of particles was formed. The wet bentonite modified by this way was used for sorption experiments.

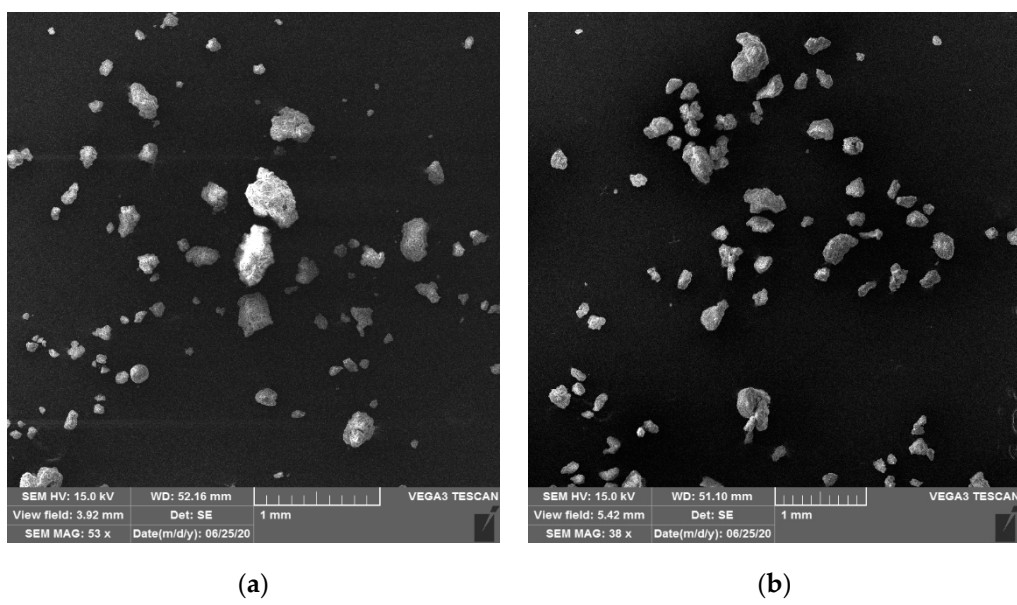

(**a**)          (**b**)

**Figure 5.** Bentonite Branany EXTRA: (**a**) before and (**b**) after hydrodynamic treatment.

### 3.3. Optimization of Laboratory Sorption Experiments

First, the height of the sorbent layer was optimized based on the maximum absorption of microparticles at the highest possible flow rate. The height of the bentonite layer was tested in the range of 2–20 cm. It was found that the height of the bentonite bed of above 13 cm that caused the column clogged after a flow of several hundred milliliters of water. On the contrary, the column with the height below 5 cm supported the too fast flow. It immediately led to the formation of cavities in the sorbent bed, which significantly reduced the efficiency of the process. For that reason, the sorbent zeolite clinoptilolite with larger and more stable particles was tested as an auxiliary support layer under the bentonite. Only the 2 cm layer of clinoptilolite placed on the perforated bottom of the column under the bentonite layer (see Experimental part, Figure 1) ensured the flow through the entire sorbent bed and prevented the formation of cavities.

During the test, 1 L of water with 50 mg of microparticles was poured into the column. Subsequently, 100 L of pure water was rinsed through the column. The best results were obtained with the hydrodynamically modified Bentonite Branany EXTRA of the 9.5 cm

bed layered on the clinoptilolite bed of 2 cm height. Afterwards, it was verified that this arrangement enabled the capture of up to 95% of microparticles and prevented the formation of cavities as well as the leaching of Branany Bentonite EXTRA. This is evident from Figure 6, which shows the images of microplastic amount at the input (a) and the output (b) of the column.

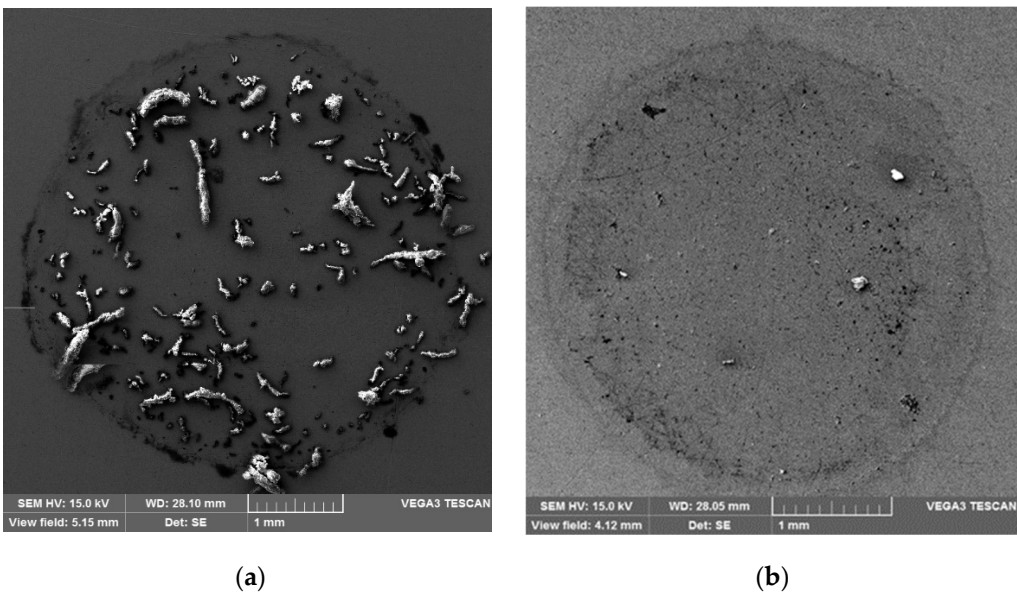

(a)                                                                                     (b)

**Figure 6.** The amount of microplastics: (**a**) at the input and (**b**) the output of the column.

Simultaneously, it was inevitable to determine the type of impurities captured at the exit of the column in order to distinguish the microplastic particles from bentonite particles, which were also weakly washed out. Therefore, parallel microplastic uptake in the sorbent was found by the elemental mapping SEM/EDX analysis (Figure 7). Figure 7a shows the analyzed area of the sample, and Figure 7b the particle type, captured microplastics (red) and sorbent particles (green and blue). The number of trapped particles per unit mass of sorbent was calculated by an image analysis. The efficiency of microplastic capture reached 90%.

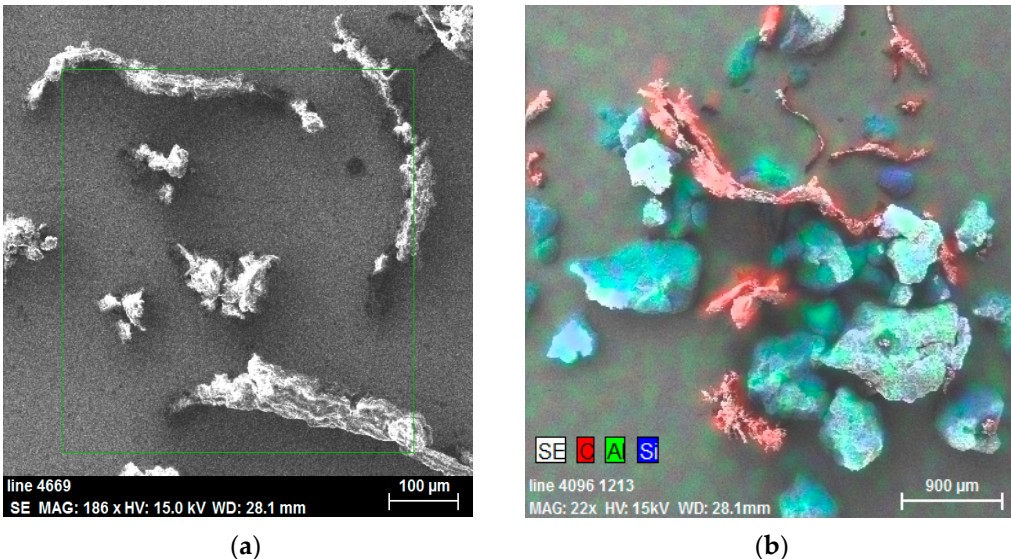

(a)                                                                                     (b)

**Figure 7.** (**a**) Analyzed area of sample; (**b**) mapping of elements C, Al and Si using SEM/EDX.

### 3.4. Design of Pilot Plant Experiments

First of all, it was necessary to develop a simple but effective treatment system for Bentonite Branany EXTRA, enabling the preparation of a sufficient amount of hydrodynamically treated sorbent required for a scale up. Thus, the Bentonite Branany EXTRA was dried at ambient temperature for approximately 7 days. Then, it was sieved by the so-called cutting (a concrete mixer with a sieve drum), which disintegrated larger clusters into a fine dust fraction of particles. The bentonite prepared this way was then soaked for 3 days, which caused swelling of the bentonite particles and their eventual disintegration. The last step was the hydraulic flotation of the bentonite, which was carried out either in a large tank of 250 L or in a cylindrical column according to the amount of bentonite being prepared.

Figure 8a shows a flow-through column which was used for a 10 kg batch of bentonite. It was placed on the sieve mesh of 2 mm and floated by water from bottom to top at the water flow rate of 4000 L/h. At the start of the floating, the finest particles of bentonite overflowed from the column; therefore, the water became strongly yellow and cloudy. The turbidity of the water gradually decreased during the process, and, at the moment of clarification, the process was terminated. The bentonite was allowed to settle due to gravity, and the water was drained from the column. The bentonite prepared this way was then placed into the adsorption column (Figure 8b). The layer of bentonite rested on the layer of pebble gravel (2–4 mm) on the support grid. This system, similar to at the laboratory scale, significantly limited the leaching of bentonite from the column.

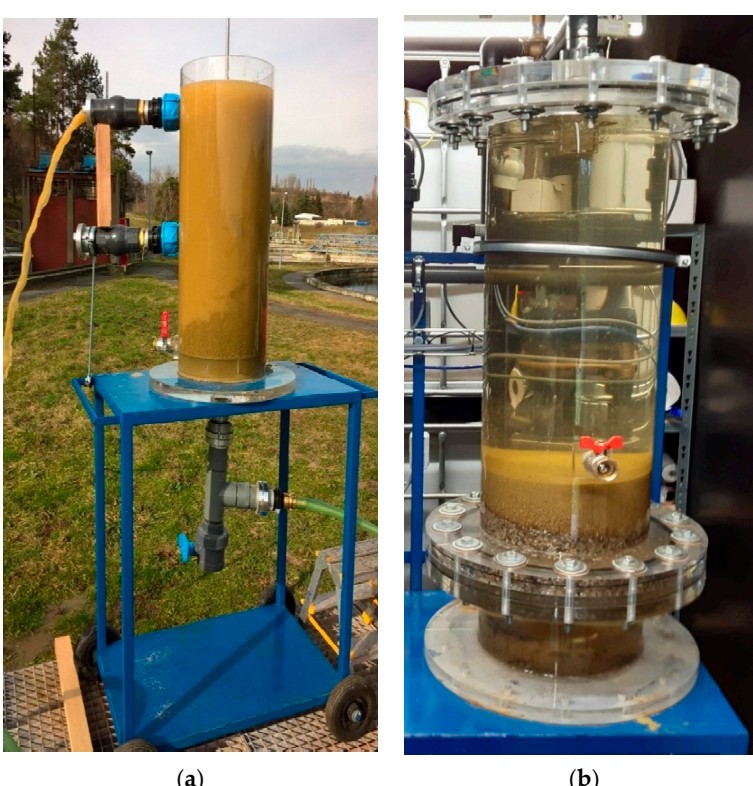

(a)                                            (b)

**Figure 8.** (**a**) Photo of a column for hydrodynamic treatment of Bentonite Branany EXTRA; (**b**) photo of developed pilot plant sorption unit for microplastics capture tested in the municipal WWTP in the Czech Republic.

The design of the newly developed sorption column apparatus was based on the cylindrical filter column with a length of 1000 mm and a diameter of 280 mm equipped with a support grid with a mesh size of 2.5 × 30 mm. In the first version, the column was made of the Plexiglas material (maximum pressure 1.3 bar) to ensure good transparency of the bentonite bed. However, the next generation of the column was made of transparent

PVC-U (non-plasticized PVC) [39], which was still sufficiently transparent and, compared to the Plexiglas material, was rated for the pressure of 4 bar. A layer of hydraulically modified bentonite with a thickness of 10–15 cm (8–15 kg) was placed on a 5–8 cm high thickness of pebble gravel. The inlet water flowed into each column by gravity through an induction flow meter and a three-way valve. The purified water left the lower part of the column equipped with a sampling valve and then through the three-way valve into the common collection pipe of the purified water. To test the efficiency of the column, the samples were taken at the inlet and outlet of the column. This column for microplastic capture was implemented in the municipal WWTP in the Czech Republic for 10 months.

*3.5. Development and Optimization of Sampling and Analytical Procedures*

An integral and very important part for the operational verification of the newly developed technology was the optimal sampling setup and mainly the design and verification of the analytical procedure for the determination of microplastics. Sampling was carried out through a tubular sampler with a system of three mesh sizes of 25, 50 and 500 μm, which were chosen on the basis of preliminary tests.

First, a visual check was carried out on the individual filters using an optical microscope with a connected digital camera. The standardized procedure of Norén [40] was used for particle identification, in which the particles on the filter were characterized regarding their size, color, shape and transparency.

Based on the preliminary measurements of the microplastic particle size and the amount and size of the found cyanobacteria, the size of the sieves used for sample sieving was chosen. The particles above 25 μm were analyzed on the filters with the largest filter mesh size of 500 μm. Thus, cyanobacteria were not considered for pilot-scale samples. The total microplastic concentration in the water taken from the waste water at each sampling point of the process was calculated based on the volume of water filtered and reported as the number of microplastics per $m^3$ of water. The methodology of particle 'concentration' evaluation was based on the methodology used in [38]. The overall procedure for characterization and identification can be found in [41].

The size distribution of individual fractions of real microplastics found in the WWTP is summarized in Table 4.

**Table 4.** Distribution of individual fractions of real microplastics.

| Filter | $C_{in}$ [1/m$^3$] | +- [1/m$^3$] | $C_{out}$ [1/m$^3$] | +- [1/m$^3$] | MP Removal [%] |
|---|---|---|---|---|---|
| 25 μm | 6800 | 550 | 650 | 70 | >90 |
| 50 μm | 6400 | 480 | 580 | 130 | >90 |
| 500 μm | 310 | 50 | 32 | 3 | >90 |

The subsamples of filtered particles (1 cm$^2$ on filters) were analyzed by Raman spectroscopy (X3) to determine the particle type. Most of the particles analyzed were above 25 μm and below 250 μm in size. Due to the significant presence of particle types other than microplastic, i.e., particles of both inorganic and organic nature in the aqueous phase, not all particles above 25 μm could be analyzed. The spectra of each measured particle were compared with a self-created database of plastic polymers with the reference spectra for the most common polymer types (PE, PP, PS, PVC, PMMA, PET, PTFE, PA, polyurethane (PU)) and categorized as microplastic (>75% correlation and/or clear agreement of characteristic peaks) or 'unknown' (<75% correlation and no agreement of characteristic peaks).

The strong interferences in the Raman spectra caused by the other materials deposited on filters were observed. The chemical, mechanical and UV effects on microplastic particles were also observed compared to the self-created database of Raman spectra. The majority of microplastic particles showed significant signs of degradation.

The qualitative analysis of microplastic particles provided interesting data. In terms of species, a total of six types of plastics were detected in the samples: PE (polyethylene), PP (polypropylene), PET (polyethylene terephthalate), PBT (polybutylene terephthalate), PES (polyester) and nylon. Figure 9 shows the dominance of PE particles in most of the collected samples (a total of 102 particles captured). A total of 7 nylon particles, 4 PP particles, 3 PES particles, 2 PET particles and 1 PBT particle were also represented. Examples of identified microplastic particles are shown in Figure 10.

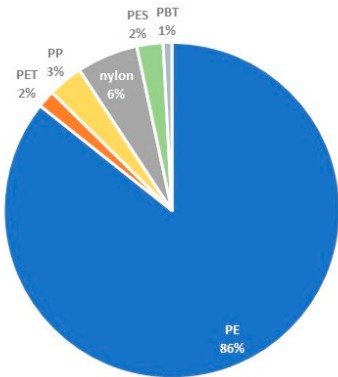

**Figure 9.** The total species composition of microplastics per 1 cm$^2$ of the metal filter evaluated from metal filters with mesh of 25 to 500 μm from all samples during the tests.

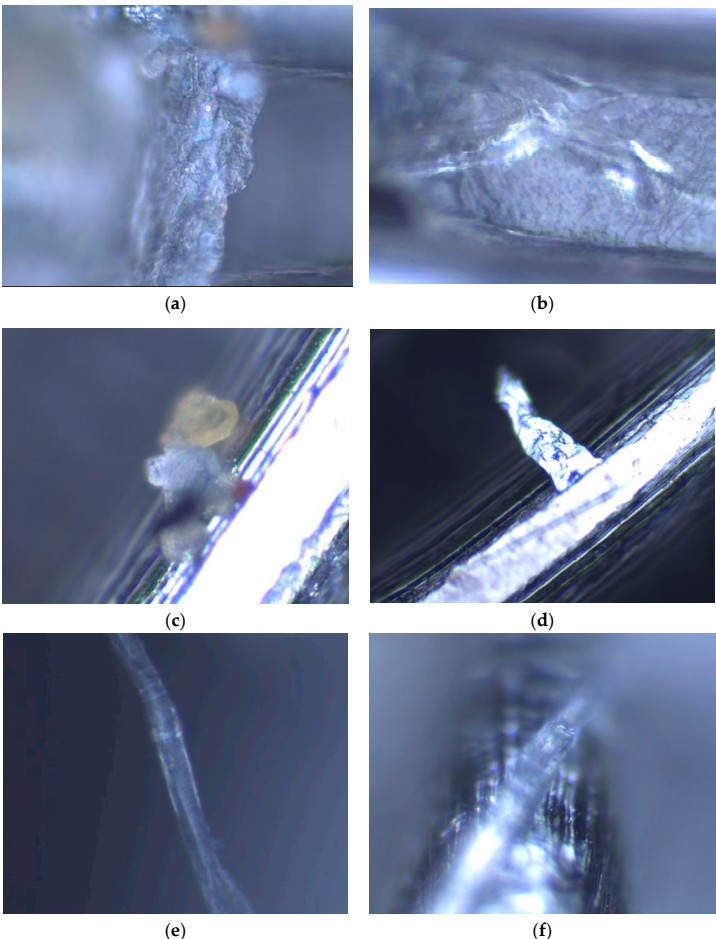

**Figure 10.** Identified microplastic particles: (**a**) PE, mesh 50 μm; (**b**) PA, mesh 25 μm; (**c**) PP, mesh 25 μm; (**d**) PES, mesh 50 μm; (**e**) PET, mesh 500 μm; (**f**) PBT, mesh 25 μm particles.

At all sampling points, the largest number of microplastic particles was captured on sieves with a mesh size of 50 and 25 μm. Detected microplastics occurred mainly in the form of flakes (PE and PP) or fibers (PA, PES, PET and PBT), while most microparticles were colorless or with a slight reddish or purple color.

The efficiency of microplastic removal by the newly designed sorption unit at the laboratory and industrial scales was evaluated based on a statistical method. The statistical results were analyzed and plotted in R-language using integrated statistical packages (StatModel). The tentative microplastics concentration in water from the Moldau River and laboratory samples were compared with a non-paired two-tailed *t*-test [35]. The particle size distributions were fitted with a one-phase decay function and bin widths of 25–50–500 μm. Further, the results from visual microplastic analysis and Raman spectroscopy were compared with Spearman's correlation at both the laboratory and industrial scales. Table 5 summarizes the concentration of microplastics at the laboratory and pilot scales regarding particle sizes of 25–750 μm.

**Table 5.** Summary of microplastic removal.

| Measurement | $C_{in}$ [1/m$^3$] | +- [1/m$^3$] | $C_{out}$ [1/m$^3$] | +- [1/m$^3$] | MP Removal [%] |
|---|---|---|---|---|---|
| Laboratory | 7800 | 600 | 570 | 90 | >93 |
| Industrial scale | 13,500 | 900 | 1100 | 150 | >91 |

The efficiency of the developed sorption column achieved 93% at the laboratory scale and 91% at the pilot scale. Nevertheless, it has to be mentioned that the removal of microplastics of larger sizes (>500 μm) was approaching 99.5% at both scales.

## 4. Conclusions

The sorption system based on Bentonite Branany and zeolite clinoptilolite for microplastic removal from water was successfully designed and tested not only at the laboratory scale but also at a designed pilot plant unit in the municipal WWTP in the Czech Republic for 10 months. To increase the sorption efficiency of Bentonite Branany, a simple, but very effective hydrodynamic treatment system was developed and successfully applied. The design sorption column system at the laboratory scale was successfully tested on all types of microplastic particles prepared by abrasion, with shapes and sizes that correspond to real microplastics found in waters. To obtain results with high reproducibility, a method for microplastic characterization based on Raman spectroscopy in combination with SEM/EDX was developed. An efficiency of 93% for microplastic removal was achieved at the laboratory scale and 91% at the pilot scale without any decrease in efficiency during the entire period in 2022.

**Author Contributions:** Conceptualization, O.S., P.D., L.W. and P.M.; methodology, M.L., L.W., P.D., P.M. and R.K.; software, M.L. and P.D.; validation, M.S., P.D., M.L., L.W., P.M. and R.K.; formal analysis, M.S.; investigation, M.S., P.D., M.L., P.M. and R.K.; resources, M.S., L.W. and O.S.; data curation, M.S., P.D., M.L., L.W., P.M. and R.K.; writing—original draft preparation, M.S, P.D, M.L., L.W., P.M., R.K. and O.S.; writing—review and editing, M.S, P.D, M.L., L.W., P.M., R.K. and O.S.; visualization, M.S., O.S.; supervision, L.W.,O.S.; project administration, L.W., P.M. and O.S.; funding acquisition, L.W., P.M. and O.S. All authors have read and agreed to the published version of the manuscript.

**Funding:** This research was funded by the Ministry of Industry and Trade of the Czech Republic, project no. FV40126.

**Data Availability Statement:** Not applicable.

**Acknowledgments:** The authors thank the Ministry of Industry and Trade of the Czech Republic for financial support (project FV40126—Advanced sorbents for the separation of microplastics and micropollutants from water).

**Conflicts of Interest:** The authors declare no conflict of interest.

## Abbreviations

| | |
|---|---|
| SEM | Scanning electron microscopy |
| PE | Polyethylene |
| PET | Polyethyleneglycol terephthalate |
| PP | Polypropylene |
| PS | Polystyrene |
| PC | Polycarbonate |
| PU | Polyurethane |
| PMMA | Polymethyl methacrylate |
| PES | Polyester |
| PBT | Polybutylene terephthalate |
| EDX | Energy-dispersive X-ray spectroscopy |
| FTIR | Fourier transform infrared spectroscop(y) |
| ATR | Attenuated total reflection |
| EG | Electric grinder |
| SP | Sand paper |
| GS | Grinding stone |
| WWTP | Wastewater treatment plant |

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
