# Peer review of "An Innovative Sorption Technology for Removing Microplastics from Wastewater"

_water, doi:10.3390/w15050892_

Round 1

Reviewer 1 Report

In a study on wastewater it cannot mark a preponderant part such as that of the microfibres produced both by domestic and industrial washing machines and by textiles in general. so I believe that all the work must be fixed by doing new analyzes and adding this very important part.

Author Response

Authors would like to thank for your feedback and the time you took to evaluate our manuscript.

In a study on wastewater it cannot mark a preponderant part such as that of the microfibres produced both by domestic and industrial washing machines and by textiles in general. so I believe that all the work must be fixed by doing new analyzes and adding this very important part.

There was probably a misunderstanding. This article is in no way aimed at determination which types of microplastics are produced in households or industry, not even with regard to the production of microplastics in washing machines (where it also depends on the type of clothing used). Determining microplastics by source is a completely different issue that is not covered in this article. The article is focused on water purification in general, i.e. from any types of microplastics, no matter where they arise, i.e. from all different sources, which reach water treatment plants. The topic of the article is their removal, no matter where they arise, so that they do not get further into nature. It is similar to the way in which the polluting agent is currently not resolved at the treatment plants which is from the household or industry (it can be the same from both), when the main task is its removal.

Reviewer 2 Report

The article aims to remove microplastics by sorption, using laboratory tests and a semi-operational sorption unit at a real wastewater treatment plan.

This article needs to be re organized, since the methods part is incomplete, this information being in the discussion of the results.

Some points need to be clarified, such as the number of tests carried out on a laboratory scale, and the respective conditions. In the topic "Optimization of laboratory sorption experiments" the authors are just checking the conditions under which it is possible to carry out the experiments, not being an optimization of the process.

 Some concrete examples that need to be improved are presented below.

·        Abstract

The abstract is very generic and should contain more information, such as:

- more details of the used sweeteners (granulometry,....);

- types of tests carried out (batch, continuous...);

- experimental conditions for laboratory and semi-operational scale tests;

- efficiency difference in the two types of tests;

- adsorption capacity of the materials.

·        Theoretical introduction

line 46 - What is the type of tertiary treatment to which the efficiencies refer?

·        Materials and methods

line 87 - "2.1. Used materials" - it would be better to separate the adsorbents from the microplastics.

line 132 - 147 - Remove repeated information. Mention the flow rate used.

The semi-operational scale procedure is not presented.

·        Results and discussion

Much of the information presented in this topic should be transferred to the materials and methods section, such as the information:

- lines 184-192

- lines 233-237

- lines 279-287

- ...

Author Response

 (Reviewer 2)

The authors thank to the Reviewer for all suggestive comments. During the revision, we have tried to clarify the ambiguous information as much as possible. The detail answers are written below.

The article aims to remove microplastics by sorption, using laboratory tests and a semi-operational sorption unit at a real wastewater treatment plan.

This article needs to be re organized, since the methods part is incomplete, this information being in the discussion of the results.

Some points need to be clarified, such as the number of tests carried out on a laboratory scale, and the respective conditions. In the topic "Optimization of laboratory sorption experiments" the authors are just checking the conditions under which it is possible to carry out the experiments, not being an optimization of the process.

 Some concrete examples that need to be improved are presented below.

Regarding laboratory test, a lot of them were carried out, in fact hundreds of tests. During this test the laboratory set-up mainly the sorbent was optimized.

  • Abstract

The abstract is very generic and should contain more information, such as:

- more details of the used sweeteners (granulometry,....);

- types of tests carried out (batch, continuous...);

- experimental conditions for laboratory and semi-operational scale tests;

- efficiency difference in the two types of tests;

- adsorption capacity of the materials.

Concerning abstract, there were also requirements that it should not be too particular and not simulate the conclusion. However, we have edited the abstract.

  • Theoretical introduction

line 46 - What is the type of tertiary treatment to which the efficiencies refer?

The efficiencies generally relate to UV, O3, chlorination, biologically active filters, disc filters and rapid sand filters. Generally, the tertiary treatment is any simple treatment, which can help with a final waste water treatment.

  • Materials and methods

line 87 - "2.1. Used materials" - it would be better to separate the adsorbents from the microplastics.

It was done.

line 132 - 147 - Remove repeated information. Mention the flow rate used.

Thank you for reminder. It was done.

The semi-operational scale procedure is not presented.

It was added according to your suggestion.

  • Results and discussion

Much of the information presented in this topic should be transferred to the materials and methods section, such as the information:

- lines 184-192

- lines 233-237

- lines 279-287

The materials and methods section contains information regarding already known methods used, or materials that can be ordered or have already been developed.

However, all information that is mentioned on lines 184-192, 233-237 and 279-287 was developed within this publication and is the result of this publication. No one has done or tried it like this before us. As part of this work, a new method was developed to obtain microplastic particles corresponding to particles found in nature. Concerning sorbent, its modification was also developed as part of this work. The semi-operational unit was also developed as a part of this work, and it is the unique unit. We also have legal protection for her, the utility model. For this reason, all this information belongs to the results and not to the materials and methods, as they would not be without our research and for which we also have legal protection. For this reason, all this information belongs to the results as they would not be without our research.

Reviewer 3 Report

The article submitted for review presents a recently frequently analyzed issue regarding the removal of microplastics from water and wastewater. In this case, a thorough analysis of possible solutions was made, as presented in the introduction. The authors presented their research, which they conducted on a laboratory and semi-technical scale. This is an additional advantage of the conducted and presented research. However, the article requires editorial correction:

1. you need to unify the units

2. make adjustments to the chapter settings.

In my opinion, this is a very important report. The more so that new methods of removing microplastics from water and sewage are still being developed.

The article interested me very much and inspired me to do further research. The article presents very interesting research that can be used to describe subsequent analyzes by a researcher in this field. 

Thank you for considering my opinion.

Author Response

The authors thank to the Reviewer for all comments, which helped us to improve the manuscript. The detail answers are written below.

The article submitted for review presents a recently frequently analyzed issue regarding the removal of microplastics from water and wastewater. In this case, a thorough analysis of possible solutions was made, as presented in the introduction. The authors presented their research, which they conducted on a laboratory and semi-technical scale. This is an additional advantage of the conducted and presented research. However, the article requires editorial correction:

  1. you need to unify the units

It has been checked and edited.

  1. make adjustments to the chapter settings.

The manuscript was edited according the suggestion and the entire manuscript has been checked.

In my opinion, this is a very important report. The more so that new methods of removing microplastics from water and sewage are still being developed.

The article interested me very much and inspired me to do further research. The article presents very interesting research that can be used to describe subsequent analyzes by a researcher in this field. 

Thank you for considering my opinion.

Reviewer 4 Report

The article is written with great care and scientific solidity. In my opinion, it is a very good publication and it shows a lot of work done. however, I have a few minor comments:

1. Please use equal spacing between chapters. You can see the differences in chapters 2.2 and 2.3

2. It would be good to standardize the notation of markings in the description of drawings (bold or not letters a, b, c, etc.)

3. Please also standardize the notation of units (mL should be) (e.g. there is a difference in lines 138 and 140). You need to read the entire article in this regard.

Apart from these minor remarks, it is a very interesting work.

The research results are discussed very well and in detail. I like that the conclusions of the research are presented in a concise way.

Moreover, the publication is a valuable source of information and forms the basis for further research.

Thank you for considering my opinion. I encourage the authors to continue working on improving the manuscript.

Author Response

The authors thank to the Reviewer for the suggestions that will allow us to improve the article.

The detail answers are written below.

The article is written with great care and scientific solidity. In my opinion, it is a very good publication and it shows a lot of work done. however, I have a few minor comments:

  1. Please use equal spacing between chapters. You can see the differences in chapters 2.2 and 2.3

The chapter settings have been adjusted.

  1. It would be good to standardize the notation of markings in the description of drawings (bold or not letters a, b, c, etc.)

It has been checked and edited throughout the manuscript.

  1. Please also standardize the notation of units (mL should be) (e.g. there is a difference in lines 138 and 140). You need to read the entire article in this regard.

Thank you for the reminder, it has been checked and unified.

Apart from these minor remarks, it is a very interesting work.

The research results are discussed very well and in detail. I like that the conclusions of the research are presented in a concise way.

Moreover, the publication is a valuable source of information and forms the basis for further research.

Thank you for considering my opinion. I encourage the authors to continue working on improving the manuscript.

Reviewer 5 Report

Whole manuscript is very well written while conclusion needs to be short and precise.

Author Response

The authors thank to the Reviewer for a very nice opinion and motivation.

Whole manuscript is very well written while conclusion needs to be short and precise.

Reviewer 6 Report

Dear Author(s),

In their study, the researchers focused on developing a technology to remove microplastics from waters (wastewater, industrial water, surface water) using zeolite and bentonite, which are economic sorbents to remove them. They found that the technology they tested was over 91% effective at removing microplastics from water. The paper is well written, has current and important data, and should be of great interest to the readers. The Introduction and others sections provide useful information for the readers. The paper has a potential to be accepted, but some important points have to be clarified or fixed before we can proceed and a positive action can be taken.

I here summarize this points:

1- If the introduction of the text provides information about the presence of microplastics in nature (sea water, beach sand, atmosphere, etc.) and food, I think it will add value to the article.

For example,

Atamanalp, M., Köktürk, M., Uçar, A., Duyar, H. A., Özdemir, S., Parlak, V., ... & Alak, G. (2021). Microplastics in tissues (brain, gill, muscle and gastrointestinal) of Mullus barbatus and Alosa immaculata. Archives of Environmental Contamination and Toxicology81, 460-469.

Desforges, J. P. W., Galbraith, M., Dangerfield, N., & Ross, P. S. (2014). Widespread distribution of microplastics in subsurface seawater in the NE Pacific Ocean. Marine Pollution Bulletin79(1-2), 94-99.

Tympa, L. E., Katsara, K., Moschou, P. N., Kenanakis, G., & Papadakis, V. M. (2021). Do microplastics enter our food chain via root vegetables? A raman based spectroscopic study on Raphanus sativus. Materials14(9), 2329.

Wright, S. L., Ulke, J., Font, A., Chan, K. L. A., & Kelly, F. J. (2020). Atmospheric microplastic deposition in an urban environment and an evaluation of transport. Environment International136, 105411.

Yabanlı, M., Yozukmaz, A., Şener, İ., & Ölmez, Ö. T. (2019). Microplastic pollution at the intersection of the Aegean and Mediterranean Seas: A study of the Datça Peninsula (Turkey). Marine Pollution Bulletin145, 47-55.

2- Information should be given about the size ranges of the final test materials (microplastics) obtained after the microplastic preparation process in the text.

3- In the text, it is stated that the effectiveness of microplastic removal with the newly designed sorption unit on laboratory and industrial scale was evaluated based on a statistical method (line 362). The statistical methods used in the study should be given under a subheading in the materials and methods section.

4- The reference list should be checked.

The content of the article in accordance with the aims of the Water.

The article is scientifically sufficient.

Keywords are well chosen so that the article can be found by indexes.

The literature has been adequately critical, current and internationally evaluated by the authors.

The language of the article is correct and clear.

The discussion part is quite comprehensive in the paper.

Tables and figures are well designed and necessary.

 Acceptable after minor revisions.

Author Response

The authors thank to the Reviewer for all comments, which helped us to improve the manuscript. The detail answers are written below.

Dear Author(s),

In their study, the researchers focused on developing a technology to remove microplastics from waters (wastewater, industrial water, surface water) using zeolite and bentonite, which are economic sorbents to remove them. They found that the technology they tested was over 91% effective at removing microplastics from water. The paper is well written, has current and important data, and should be of great interest to the readers. The Introduction and others sections provide useful information for the readers. The paper has a potential to be accepted, but some important points have to be clarified or fixed before we can proceed and a positive action can be taken.

I here summarize this points:

1- If the introduction of the text provides information about the presence of microplastics in nature (sea water, beach sand, atmosphere, etc.) and food, I think it will add value to the article.

For example,

Atamanalp, M., Köktürk, M., Uçar, A., Duyar, H. A., Özdemir, S., Parlak, V., ... & Alak, G. (2021). Microplastics in tissues (brain, gill, muscle and gastrointestinal) of Mullus barbatus and Alosa immaculata. Archives of Environmental Contamination and Toxicology81, 460-469.

Desforges, J. P. W., Galbraith, M., Dangerfield, N., & Ross, P. S. (2014). Widespread distribution of microplastics in subsurface seawater in the NE Pacific Ocean. Marine Pollution Bulletin79(1-2), 94-99.

Wright, S. L., Ulke, J., Font, A., Chan, K. L. A., & Kelly, F. J. (2020). Atmospheric microplastic deposition in an urban environment and an evaluation of transport. Environment International136, 105411.

Yabanlı, M., Yozukmaz, A., Şener, İ., & Ölmez, Ö. T. (2019). Microplastic pollution at the intersection of the Aegean and Mediterranean Seas: A study of the Datça Peninsula (Turkey). Marine Pollution Bulletin145, 47-55.

Thank you for your suggestion, we added more information about the presence of microplastics according your proposal.

Four high production volume polymers applied in plastic (polyethylene terephthalate, polyethylene, polystyrene derivates as a expanded polystyrene, acetonitrile butadiene styrene etc.) were identified and quantified for the first time in blood. [Leslie  et. al]

Atamanalp, M., Köktürk, M., Uçar, A., Duyar, H. A., Özdemir, S., Parlak, V., ... & Alak, G. (2021). Microplastics in tissues (brain, gill, muscle and gastrointestinal) of Mullus barbatus and Alosa immaculata. Archives of Environmental Contamination and Toxicology, 81, 460-469.

Tympa, L. E., Katsara, K., Moschou, P. N., Kenanakis, G., & Papadakis, V. M. (2021). Do microplastics enter our food chain via root vegetables? A raman based spectroscopic study on Raphanus sativus. Materials, 14(9), 2329.

Leslie, H. A., Van Velzen, M. J. M., Brandsma, S. H., Vethaak, A. D., Garcia-vallejo, J. J., & Lamoree, M. H. (2022). Discovery and quantification of plastic particle pollution in human blood. Environment International, 163, 1-8. [107199]. https://doi.org/10.1016/j.envint.2022.107199

2- Information should be given about the size ranges of the final test materials (microplastics) obtained after the microplastic preparation process in the text.

Comments to the size distribution of prepared and real microplastics were added to the manuscript.

3- In the text, it is stated that the effectiveness of microplastic removal with the newly designed sorption unit on laboratory and industrial scale was evaluated based on a statistical method (line 362). The statistical methods used in the study should be given under a subheading in the materials and methods section.

Thank you for the reminder, it has been added to the text.

4- The reference list should be checked. – It has been done.

The content of the article in accordance with the aims of the Water.

The article is scientifically sufficient.

Keywords are well chosen so that the article can be found by indexes.

The literature has been adequately critical, current and internationally evaluated by the authors.

The language of the article is correct and clear.

The discussion part is quite comprehensive in the paper.

Tables and figures are well designed and necessary.

 Acceptable after minor revisions.

Reviewer 7 Report

The manuscript is interesting; however, it has some problems that are listed below:

1) Abstract is not so attractive and the authors should rewrite it. For example, Lines 23-25 refer to the methods and shouldn’t be added in the abstract. Also, the results are poorly highlighted in the abstract.

2) Introduction section (Lines 49-58). Several methods of microplastics are mentioned, however, the efficiencies are not shown. Please added the removal percentage to compare them. The authors should rewrite this part of the Introduction section (Lines 48-78), it is confusing and does not have a logical sense.

3) More information about the microplastics should be provided, such as type, surface area, format, average size, etc. How the removal of microplastics was quantified? As the authors dosed the microplastic by weight, how the number of particles was calculated?

4) Microplastic suspension. How was prepared the testing water (type, pH, and alkalinity)? Please, add more details about it. Where is the information about the wastewater?

5) Please add the average size of each adsorbent. (Table 2). It will provide information about the pore size and help to analyze the time of method is occurring in your experiments. For example, physical filtration can also be contributing to microplastic removal. Also, add the BET analysis of the microplastics.

6) The manuscript should be revised and improved. It is too confusing. Please split the section preliminary section (bench test) and validation (pilot test), they are not clear in the materials and method section. The results should be refined. The discussion should be improved, there is no comparison with other studies. There are many figures and some of them could be removed.

Author Response

Authors would like to thank you for your feedback and the time you took to evaluate our manuscript. We have revised the text according to your suggestions and feel that it helped clarify and improve our paper.

The manuscript is interesting; however, it has some problems that are listed below:

1) Abstract is not so attractive and the authors should rewrite it. For example, Lines 23-25 refer to the methods and shouldn’t be added in the abstract. Also, the results are poorly highlighted in the abstract.

Changes to the abstract to meet your comments have been done. Thank you.

2) Introduction section (Lines 49-58). Several methods of microplastics are mentioned, however, the efficiencies are not shown. Please added the removal percentage to compare them. The authors should rewrite this part of the Introduction section (Lines 48-78), it is confusing and does not have a logical sense.

More details on microplastics removal using various techniques have been added to the manuscript, including their removal efficiency and have been commented on/compared.

3) More information about the microplastics should be provided, such as type, surface area, format, average size, etc. How the removal of microplastics was quantified? As the authors dosed the microplastic by weight, how the number of particles was calculated?

Detailed information has been added to the manuscript.

The size distribution of individual fractions of real microplastics found at WWTP is summarized in following table:

Table 2. Distribution of individual fractions of real microplastics

Filter

Cin

[1/m3]

+-

[1/m3]

Cout

[1/m3]

+-

[1/m3]

MP removal

 [%]

25 µm

6800

550

650

70

>90

50 µm

6400

480

580

130

>90

500 µm

310

50

32

3

>90

4) Microplastic suspension. How was prepared the testing water (type, pH, and alkalinity)? Please, add more details about it. Where is the information about the wastewater?

Tap water (let the water run of the tap 5 minutes) has been used to simulate potable water. The pH has been tested once a week with small differences found (6.6+-0.2). The salinity has not been tested.

5) Please add the average size of each adsorbent. (Table 2). It will provide information about the pore size and help to analyze the time of method is occurring in your experiments. For example, physical filtration can also be contributing to microplastic removal. Also, add the BET analysis of the microplastics.

The treatment of individual bentonite batches lead to formation of different size distributions (obtained from picture analysis from SEM data). Generally the largest particles (>5000 µm) formed less then (10-15%) m/m of the population and sedimented fast after filling the column with water and bentonine. The second most abundant fraction (by weight) was between 750 and 5000 µm (55-70 %). In course of time a small part of this fraction was fragmented by action of water and formed particles mainly with size under 500 µm. Another fraction observed was < 500 µm with mass fraction of ca 15-20%.

A small fraction (<10-15% m/m) under 100 µm was flushed as described in publication.

Thank you for reminder. The information about sorbent properties as size distribution of the bentonite has been added to the manuscript. No data has been obtained from BET analysis of both real and prepared microplastics.

6) The manuscript should be revised and improved. It is too confusing. Please split the section preliminary section (bench test) and validation (pilot test), they are not clear in the materials and method section. The results should be refined. The discussion should be improved, there is no comparison with other studies. There are many figures and some of them could be removed.

Both, preliminary section and validation tests were splitted according the suggestion and comparison with the other studies were added into the manuscript; in Introduction as well, in Results and Disscussion.

Round 2

Reviewer 1 Report

The paper is interesting and I recommend it for the publication 

Author Response

The authors thank you for your opinion and recommendations. The English was checked by a native speaker and hopefully it will be fine now.

Reviewer 2 Report

The requested changes were made and the explanations mentioned by the authors were accepted. 

Author Response

The authors thank you for your opinion.

Reviewer 7 Report

The article's quality has improved significantly.

Author Response

The authors thank you for your opinion.